# Inflammatory Bowel Disease: Are Symptoms and Diet Linked?

**DOI:** 10.3390/nu12102975

**Published:** 2020-09-29

**Authors:** Hannah Morton, Kevin C. Pedley, Robin J. C. Stewart, Jane Coad

**Affiliations:** 1School of Food & Advanced Technology, College of Sciences, Massey University, Palmerston North 4442, New Zealand; j.coad@massey.ac.nz; 2School of Health Sciences, College of Health, Massey University, Palmerston North 4442, New Zealand; k.c.pedley@massey.ac.nz; 3School of Applied Science, Universal College of Learning, Palmerston North 4442, New Zealand; b.stewart@ucol.ac.nz

**Keywords:** inflammatory bowel disease, Crohn’s disease, ulcerative colitis, diet, symptom, exclusion

## Abstract

New Zealand (NZ) has one of the world’s highest incidence rates of Inflammatory Bowel Disease (IBD), a group of chronic inflammatory conditions that affect the gastrointestinal tract. Patients with IBD often believe certain foods influence their disease symptoms and consequently may alter their diet considerably. The objective of this study was to determine foods, additives, and cooking methods (dietary elements) that NZ IBD patients identify in the onset, exacerbation, or reduction of their symptoms. A total of 233 participants completed a self-administered questionnaire concerning symptom behaviour in association with 142 dietary elements. Symptom onset and symptom exacerbation were associated with dietary elements by 55% (128) and 70% (164) of all IBD participants, respectively. Fruit and vegetables were most frequently identified, with dairy products, gluten-containing bread, and foods with a high fat content also considered deleterious. Of all IBD participants, 35% (82) associated symptom reduction with dietary elements. The identified foods were typically low in fibre, saturated fatty acids, and easily digestible. No statistically significant differences were seen between the type or number of dietary elements and disease subtype or recent disease activity. The association between diet and symptoms in patients with IBD and the mechanism(s) involved warrant further research and may lead to the development of IBD specific dietary guidelines.

## 1. Introduction

Crohn’s disease (CD) and ulcerative colitis (UC), collectively termed inflammatory bowel disease (IBD), are chronic inflammatory conditions that affect the gastrointestinal (GI) tract [1]. In UC only the colon is affected, whereas CD can affect any region of the GI tract. The prevalence in North America, Europe, and Oceania is estimated to exceed 0.3%, while newly industrialised countries with low prevalence rates are seeing a rapid increase in incidence [2,3]. The cause of IBD is currently unknown, although evidence suggests that genetic susceptibility, immune system dysregulation, and environmental factors are all involved [4]. Symptoms of IBD typically include diarrhoea and abdominal pain [5], and patients with CD are at greater risk of nutrient deficiency owing to increased requirements, inadequate dietary intake or composition, and malabsorption due to chronic inflammation and resection history [6]. Management of IBD involves the use of anti-inflammatory drugs, corticosteroids, and biologics to induce and maintain remission, and surgery for non-responsive patients and complications associated with disease progression [7].

Diet has long been implicated in the aetiology of IBD, a theory reinforced by the reduction of inflammation in response to enteral nutrition (EN) as well as its high efficacy for remission induction in patients with CD [8]. Early observational studies investigated the possible involvement of select dietary components including intake of sweets and pastries [9] and breakfast habits [10,11], yet many of these studies are fraught with methodological problems and the findings are often un-replicated [12]. Large-scale epidemiological studies are better suited to investigating dietary patterns and the findings may provide aetiological clues; however, they are difficult to translate into practical dietary guidelines that patients with IBD frequently request.

More than half of patients with IBD believe their symptoms are induced or exacerbated by specific foods [13]. Commonly identified foods include fruit and vegetables, dairy products, spicy foods, processed foods, nuts and seeds, alcohol, and foods with a high fat content [14,15,16,17]. Promising results evidenced by prolonged remission have been observed in response to traditional exclusion and re-challenge trials [18,19,20]; however, there is uncertainty around the underlying mechanism(s) responsible. Investigation of GI symptoms in patients with IBD has demonstrated the coexistence of functional GI disorders (FGID), such as irritable bowel syndrome (IBS) or functional constipation [21]. An exclusion diet effective for reducing functional GI symptoms in patients with IBS is the FODMAP diet that limits fermentable oligosaccharides, disaccharides, monosaccharides, and polyols. Recently, patients with IBD reporting functional GI symptoms have also been shown to benefit from this diet [22,23,24]. Many other forms of exclusion diet have been trialled during the last half century such as the specific carbohydrate diet [25], Crohn’s disease exclusion diet [26], IBD anti-inflammatory diet [27], and low residue diet [28]. Currently the evidence available is insufficient to classify dietary interventions of this nature as beneficial in the induction or maintenance of remission in IBD [29].

It is evident that dietary modification could have a role in the management of IBD symptoms. One difficulty in advancing this concept is a lack of knowledge around the scope of dietary elements implicated and the physiological processes involved. A question that needs to be answered is whether a common factor or pattern exists among dietary elements excluded by patients with IBD, as well as those they favour. In the present observational study, a self-administered retrospective questionnaire was used to investigate the extent that NZ IBD patients associate disease symptoms with their diet. The objective of this study was to determine foods, additives, and cooking methods (dietary elements) identified in the onset, exacerbation, or reduction of IBD disease symptoms.

## 2. Materials and Methods

### 2.1. Participants

Patients diagnosed with IBD were invited to complete a self-administered retrospective questionnaire on associations between foods and IBD symptoms. Advertisements were placed in gastroenterology clinics and at community IBD support organisations located throughout NZ. The inclusion criteria were 16 years or older; diagnosis of CD, UC, or IBD-unclassified (IBDU); and predominantly living in NZ for the previous two years. Screening of potential participants was undertaken by phone or in person by the primary investigator. Individuals were asked to confirm the form of IBD they have been diagnosed with, that the diagnosis had been made by a gastroenterologist or similarly qualified medical professional, and how the diagnosis was made (e.g., one or more of endoscopic, surgical, radiological and biochemical investigation). A minimum of two years living in NZ was considered an adequate time period for immigrants to be exposed to the food items listed in the questionnaire, particularly seasonal fruit and vegetables.

### 2.2. Questionnaire

A review of the literature was undertaken to identify foods implicated in the onset, exacerbation and reduction of IBD symptoms. Additional foods were identified from the adult national nutrition survey to ensure inclusion of foods commonly consumed in NZ [30]. The questionnaire contained 142 dietary elements comprising 118 foods, 13 beverages, 7 additives, and 4 cooking methods (Appendix A). The dietary elements were listed in columns and arranged into 15 groups: fruit, vegetables, nuts/seeds/dried fruit, grains, bread, cereals, miscellaneous, dairy products, meat, sauces, spreads, sweets/snacks/beverages, additives, and cooking methods. The complete list of dietary elements was provided in triplicate, one for each IBD symptom association; onset, worsening (exacerbation), and reduction. Participants indicated an association between a food element and their IBD symptoms and could provide additional information in open-ended question fields. Questionnaires were provided in hardcopy or via a secure online survey platform (surveymonkey.com).

### 2.3. Ethics

The protocol for this study was approved by the Massey University Human Ethics Committee: Southern A, Palmerston North, New Zealand (MUHEC Reference 13/58). All subjects were verbally screened and understood that by completing the questionnaire they were implicitly consenting to inclusion in the study.

### 2.4. Statistical Analysis

Statistical analysis was performed using JASP [31]. Statistical differences between continuous demographic variables were assessed by 1-way ANOVA. Associations between disease subtype and categorical demographic variables including gender, family history of IBD, and reported disease activity in the last 12 months were assessed by Chi-square test. Chi-square tests were also used to identify associations between dietary elements, food groups, and fruit and vegetable FODMAP content (Monash University FODMAP diet mobile application, version 3.0.3) with reported symptom effect. Associations between disease subtype (CD and UC) were also performed. T-tests were used to compare differences between participants who reported active disease in the previous 12 months versus those who reported inactive disease. Significance was set at *p* < 0.05.

Comparison with disease subtype IBDU was not performed due to an insufficient number of participants with this diagnosis. When comparing FODMAP content with reported symptom effect, lentils were excluded from FODMAP analyses as the FODMAP content is considered medium.

## 3. Results

### 3.1. Baseline Participant Characteristics

Two hundred and fifty-six individuals registered interest in the study. Two were excluded due to not meeting the inclusion criteria. One participant withdrew during the study citing difficulties completing the questionnaire. A total of 20 participants did not complete the questionnaire and were unable to be contacted further. Of the 233 participants, 71% were female. Disease subtype comprised CD 63%, UC 32%, and IBDU 5%. The mean age of participants was 40 years with no significant differences between the three groups (*p* = 0.105). The mean reported age at diagnosis was 29 years and differed significantly between participants with CD and participants with IBDU, 28 and 38 years, respectively (*p* = 0.015). Mean disease duration was similar in the three groups, 11.6 years for participants with CD, 9.9 for participants with UC, and 10.7 years for participants with IBDU (*p* = 0.510). A family history of IBD was reported by 29% of participants. The characteristics are given in Table 1.

### 3.2. Onset of Symptoms

An association between one or more dietary elements and symptom onset was reported by 128 (55%) participants. The mean number of dietary elements reported to initiate the onset of symptoms was 67.0 ± 5.1 for all participants, 68.9 ± 6.6 for participants with CD, and 58.4 ± 8.2 for participants with UC. The difference between CD and UC was not statistically significant (*p* = 0.330). Seven (5.5%) participants identified a single dietary element, nine (7.0%) participants identified between two and five dietary elements, and six or more dietary elements were identified by 112 (87.5%) participants. The number of dietary elements associated with symptom onset was not significantly different between participants who reported active disease in the previous 12 months and those who reported inactive disease. Significantly more dietary elements were associated with symptom onset than symptom exacerbation (*p* = 0.006).

The dietary elements most frequently identified in symptom onset were deep fried cooking method (73%), apple (63%), full-grain bread (62%), ice cream (61%), wheat (59%), kiwifruit (59%), corn (59%), cabbage (57%), onion (57%), and fried cooking method (57%). Appendix A gives the 25 most frequently identified dietary elements in symptom onset. The most frequently identified dietary elements according to group were vegetables (84%), beverages (80%), fruits (77%), dairy products (73%), cooking methods (73%), and sweets/snacks (70%).

### 3.3. Exacerbation of Symptoms

An association between one or more dietary elements and symptom exacerbation was reported by 164 (70%) participants. Overall, the mean number of dietary elements reported to exacerbate symptoms was 48.7 ± 4.3 for each participant. When analysed by disease subtype, there was no significant difference in the mean number of dietary elements reported to exacerbate symptoms between participants with CD and UC (43.8 ± 5.2 vs. 51.0 ± 7.5, *p* = 0.429).

A single dietary element was identified by 6 (3.7%) participants, 20 (12.2%) participants identified between two and five dietary elements, and six or more dietary elements were identified by 119 (72.6%) participants. Nineteen (11.5%) participants specified one or more dietary elements in the other foods section of the questionnaire and did not identify any listed dietary elements. The number of dietary elements associated with symptom exacerbation was not significantly different between participants who reported active disease in the previous 12 months and those who reported inactive disease.

The dietary elements most frequently identified in symptom exacerbation were deep fried cooking method (60%); full-grain bread (55%); alcohol (49%); cabbage (48%); chilli sauce (48%); ice cream (47%); wheat (46%); wholemeal bread (46%); muesli (46%); and onion, chilli, and whole cow’s milk (45% each). Appendix A gives the 25 most frequently identified dietary elements in symptom exacerbation. The most frequently identified dietary elements according to group were vegetables (80%), beverages (79%), cooking methods (70%), bread (68%), fruit (67%), and dairy products (66%).

### 3.4. Reduction of Symptoms

An association between one or more dietary elements and symptom reduction was reported by 82 (35%) participants. The mean number of dietary elements reported to reduce symptoms was 19.5 ± 4.5 for each participant. Participants with UC identified almost twice as many dietary elements than participants with CD, although the difference was not statistically significant (28.5 vs. 13.8, *p* = 0.132). A single dietary element was identified by 10 (12.2%) participants, 28 (34.1%) participants identified between two and five dietary elements, and six or more dietary elements were identified by 34 (41.5%) participants. Ten (12.2%) participants specified one of more dietary elements in the other foods section of the questionnaire and did not identify any listed dietary elements. The number of dietary elements associated with symptom reduction was not significantly different between participants who reported active disease in the previous 12 months and those who reported inactive disease. Significantly fewer dietary elements were associated with symptom reduction than symptom onset (*p* < 0.001) or symptom exacerbation (*p* < 0.001).

The dietary elements most frequently identified in symptom reduction were banana (39%), rice (35%), white bread (33%), chicken (33%), oily fish (28%), non-oily fish (27%), eggs (26%), pumpkin (24%), yoghurt (24%), and baked cooking method (24%). Appendix A gives the 20 most frequently identified dietary elements in symptom reduction. The most frequently identified dietary elements according to group were fruit (53%), bread (51%), vegetables (49%), grains (47%), dairy products (46%), and meat (46%).

### 3.5. Other Responses

Additional information provided by participants demonstrated that the onset or exacerbation of symptoms were also associated with spicy foods (11%), foods with a high fat content (8%), takeaways or deep-fried foods (5%), raw fruit or vegetables (4%), processed foods (4%), fibrous foods (4%), foods with a high sugar content (4%), foods with a high FODMAP content (4%), and barbequed foods (1%). Other dietary elements reported in association with the reduction of symptoms were plain or bland foods (9%), soft or pureed foods (6%), boiled or mashed potato (5%), smoothies (4%), ginger beer or ale (4%), white bread or toast (4%), steamed foods (4%), yoghurt (2%), low FODMAP foods (2%), porridge (2%), and turmeric (2%).

### 3.6. FODMAPS

Fruits with a high FODMAP content were associated with symptom onset (*p* = 0.004) and symptom exacerbation (*p* = 0.009) significantly more than fruit with a low FODMAP content. Low FODMAP vegetables were associated with symptom onset (*p* < 0.001) and symptom exacerbation (*p* < 0.001) significantly more than high FODMAP vegetables. No difference was observed between the FODMAP content of vegetables or fruit associated with symptom reduction. There was also no difference between the FODMAP content of vegetables or fruit associated with symptom onset, exacerbation, or reduction between participants with CD and UC.

## 4. Discussion

### 4.1. Onset and Exacerbation of Symptoms

In the present study, 55% of participants reported an association between dietary elements and the onset of their IBD symptoms, consistent with the range of 33–68% observed in other studies [15,17,32,33,34]. Moreover, within the range seen in other studies (44–76.5%) [32,34,35,36,37,38] a greater percentage of participants (70%) reported an association between diet and symptom exacerbation.

The majority of participants identified six or more dietary elements in the onset or exacerbation of symptoms, while less than 13% of participants identified between two and five, and less than 6% identified a single item. These findings were higher than anticipated, although this is likely a reflection of the number of dietary elements evaluated and a tendency of some participants to indicate all items within a food group, notably vegetables. Surprisingly, there was no difference in the number of dietary elements identified overall or between participants with CD and UC, a finding that contradicts with observations of greater food avoidance by patients with CD [35,39,40].

Likewise, no difference was seen in the number of dietary elements identified by participants who reported active disease in the previous 12 months and those who reported inactive disease. As active disease is associated with significantly greater food avoidance [39,40,41], it was expected that the recall of dietary elements associated with symptoms would be heightened in participants who reported active disease and diminished in patients reporting extended remission.

### 4.2. Fruit and Vegetables

Patients with IBD often demonstrate an aversion for fruit and vegetables, a behaviour believed to be due to the fibre content and the effects they may experience following consumption including bloating, abdominal pain, and diarrhoea [42]. Accordingly, participants in this study frequently identified fruit and vegetables in the initiation or exacerbation of their IBD symptoms, a commonly reported finding [33,41,43] along with highly fibrous foods [14,44].

In Zallot et al.’s [15] study on dietary beliefs and behaviour, 47.5% of participants perceived raw vegetables to be a relapse risk factor. Similarly, a study on Canadian dietary habits reported avoidance of raw vegetables in 46% of participants [16]. This was not observed in the present study as the most frequently identified vegetables, including corn, cabbage, onion, chilli, broccoli, and garlic, are generally cooked before consumption. However, a small proportion of participants remarked that raw fruit and vegetables are problematic. A trend towards cruciferous [15] and brassica [39] vegetables has also been described; again, this was not supported by our findings with only two of the most identified vegetables meeting this criterion.

Research suggests more than a third of patients with IBD experience functional GI symptoms compatible with IBS [45] and that a low FODMAP diet can significantly reduce functional GI symptoms [22,23]. In fact, many of the foods frequently identified in this study have also been identified by patients with IBS as problematic including chilli, spices, onion, cabbage, alcohol, fried and fatty foods, and caffeine [46,47]. This might indicate an association between the FODMAP content of fruit and vegetables and symptom response. This hypothesis was supported by the significant association between fruits with a high FODMAP content and both symptom onset and exacerbation; however, for vegetables the opposite was observed. It is unclear why vegetables with a low FODMAP content were associated with symptoms. This could indicate a low incidence of functional GI symptoms experienced by this cohort, or alternatively, another characteristic of vegetables could affect symptoms to a greater extent than the FODMAP content.

In line with similar studies, fruit were identified in symptom onset and exacerbation less often than vegetables. The diversity of fruit identified was also less than vegetables, making it difficult to identify a common factor. Citrus fruits have previously been singled out as being problematic [32,38,43,48], an observation not evident in this study with only one citrus fruit, orange, amongst the most frequently identified fruits. Considering each fruit individually, it is unclear how apple and orange could elicit GI irritation. Whereas the seed content of tomatoes and kiwifruit may be problematic as patients with IBD largely deem seeds to be deleterious. Additionally, kiwifruit may cause diarrhoea as a consequence of its laxative effect [49].

### 4.3. Dairy Products

Avoidance of dairy products is commonplace in patients with IBD. Of the nine dairy products evaluated in this study, ice cream and cream were most frequently identified by all participants. Participants with CD also identified hard cheese and fruit yoghurt in association with symptom onset, and both whole and reduced fat cow’s milk were identified by participants with UC. In some participants, the association between symptoms and dairy products could possibly be attributable to lactose intolerance, a term used to describe the onset of GI symptoms in response to lactose consumption, usually due to lactase deficiency induced malabsorption [50]. A degree of overlap exists between symptoms of IBD and lactose intolerance including bloating, abdominal pain, and diarrhoea [51]; thus, participants may be unaware that these symptoms can occur independently of IBD or be unable to distinguish between symptoms.

An NZ study investigating the effects of dairy products in 165 patients with CD found that the worsening of symptoms correlated with fat content, with the most problematic dairy products being cream, cheese, ice-cream, and standard milk [52]. The findings of this study support the notion that fat content could be a key component in symptom onset or exacerbation, although participants with CD also frequently identified fruit yoghurt which usually has a low-fat content. Further, butter was not identified as often as some dairy products with significantly lower fat content, a mutual finding that somewhat undermines the fat content theory or could equally reflect that consumption quantities are usually inadequate to elicit an effect [52].

### 4.4. Gluten

An unexpected finding was the association between bread and the onset and exacerbation of symptoms. Of the five types of bread listed in the questionnaire, only gluten-free bread was well tolerated, a finding also seen in Triggs et al.’s [53] study of food intolerances. The coexistence of coeliac disease or non-coeliac gluten sensitivity are possible reasons for these findings and are further supported by the frequent reporting of wheat and wheat-based cereal in this study. However, the likelihood of undiagnosed coeliac disease in a patient with IBD is low despite the relative risk of coeliac disease being almost four times higher in patients with IBD [54]. Moreover, taking into account the estimated Oceanic coeliac disease prevalence of 0.8% [55], this would suggest a prevalence of only 3.2% in the NZ IBD population. Non-coeliac gluten sensitivity, on the other hand, affects approximately 30% of patients with IBD [56]. Significant benefit to patients with IBD has also been demonstrated in response to a gluten free diet trial with 66% of the 314 patients reporting symptom improvement and reduced flare frequency and severity in 38% [57].

### 4.5. Other

Of interest are the strong associations that participants have with cooking methods. Deep frying was the dietary element most frequently identified in symptom onset and exacerbation, ahead of foods, and frying was also frequently identified. Both cooking methods generally involve the addition of fat as a cooking medium, and thus, they are suggestive of deleterious effects in response to the ingestion of foods with a high fat content. The study was not designed to explore the perceived effects of macronutrients; however, from the other responses received, 8% of participants specified foods with a high fat content, and 5% specified takeaways or deep-fried foods. These findings are in agreement with previous and current studies where patients with IBD have identified fast food [35,48], fatty food [15,17,34,36,37], fried and fatty foods [14], deep-fried and fatty food [16,39], oily food [38], and fatty meats [58] in symptom onset or exacerbation. Furthermore, evaluation of dietary intake changes in over 4200 families in Japan over two decades found positive associations between increasing CD incidence and total fat, animal fat, and omega-6 fatty acids [59]. Whether GI irritation is evoked in response to the type or content of fat ingested remains to be elucidated.

A large percentage of participants identified alcohol, fruit juice, chilli sauce, and coffee in the onset and exacerbation of symptoms. Given that alcohol can cause GI symptoms in healthy individuals, it was unsurprising that wine and beer specifically, and alcohol in general, were frequently identified. Fruit juice was likely identified for the same reason as fruit and vegetables, and the capsaicin content could explain why chilli sauce, and perhaps spicy foods also, are frequently deemed problematic [40]. Unlike most studies, the perceived effects of coffee and tea were investigated individually, where coffee was frequently associated with symptom onset and exacerbation, and black tea was well tolerated. Generally, tea and coffee appear to be problematic as evidenced by their association with increased risk of relapse [15], negative effects on symptoms [48], significantly reduced consumption in CD patients experiencing active disease [60], and overall avoidance [16,58]. However, taking into account de Vries et al.’s [32] observation that tea is the second most symptom improving food among 294 patients with IBD, and the findings of the present study, coffee and tea should be considered separately when investigating their purported effects on IBD symptoms.

### 4.6. Reduction of Symptoms

The view that diet can directly influence IBD symptoms is most convincingly demonstrated by the ability of EN to ameliorate inflammation. Although not definitive, EN is thought to reduce inflammation by downregulating proinflammatory cytokine levels, facilitating mucosal healing and restoring GI barrier function, and improving nutritional status [61]. In the current study, less than half of the participants (35%) reported an association between dietary elements and symptom reduction. Dietary elements identified in both the current study and previous studies as being associated with symptom improvement include bananas, chicken, rice, white bread, oily fish, white fish, yoghurt, gluten free foods, and herbal tea [14,32,38,48,53,62]. Cooking foods by baking or grilling were also frequently identified in the current study. Additionally, participants specified that they associate symptom reduction with plain or bland foods, soft or pureed foods, boiled or mashed potato, smoothies, ginger beer or ale, and steamed foods.

Reduction of inflammation and ultimately mucosal healing is the goal of IBD management. The anti-inflammatory effects of omega-3 fatty acids are well established, particularly for improving health outcomes in those affected by chronic conditions [63]. While the evidence from available studies is considered inadequate to make recommendations, omega-3 fatty acid supplementation in patients with IBD has been shown to reduce disease activity, extend remission, improve endoscopic scores, and decrease levels of inflammatory markers [64,65]. As oily and non-oily fish were frequently identified by participants, and avocado by participants with UC, the reported reduction of symptoms could be facilitated by omega-3 or monounsaturated fatty acid intake. This is further supported by a recent dietary belief and behaviour study where 2% of patients with inactive UC reported consumption of oily fish or omega-3 supplements to prevent relapse [17].

Theoretically, attributes of the dietary elements identified in symptom onset and exacerbation are deemed potentially harmful by the immune system and consequently trigger an inflammatory response. Thus, the content of suspected irritants may be lower in dietary elements identified in symptom reduction. Accordingly, their overall fibre content is low; dairy product number is limited and low in lactose; saturated fatty acid levels are minimal; wholegrains, spicy foods, alcohol, and caffeine are notably absent; gluten is avoided, and the cooking methods require little or no added fat. The extent of these dietary restrictions possibly equates to a significantly reduced antigenic load and provision of nutrients in a form that eases digestion requirements, elements also believed to be of importance to EN [61].

In agreement with Triggs et al. [53], the views regarding diet and IBD symptoms demonstrate that individual foods cannot be classified as either detrimental or beneficial for all patients with IBD. This indicates that like other aspects of IBD, including disease trajectory and response to therapy, patients respond to dietary elements in a heterogenous fashion. Whole cow’s milk is a prime example with 57% of participants with UC reporting an association with symptom exacerbation, and 30% with symptom reduction. Similarly, 39% of participants with CD identified orange in symptom exacerbation, and 16% in symptom reduction. Other instances may be attributable to changes in food properties following preparation and/or cooking. For example, apple was identified in symptom onset by 66% of participants with CD, and in symptom reduction by 18%. In the case that mechanical processing and digestibility are directly correlated to symptoms, it is plausible that the effect of raw apple could be distinct to that of peeled and stewed apple.

### 4.7. Dietary Modification

According to studies on dietary beliefs and behaviours, 56–90% [33,34,35,39,41,48] of patients modify their diet following their diagnosis of IBD. The most recent of these studies reported that foods rich in fibre, fruit and vegetables, and grains are the most modified foods [41], putting patients at risk of inadequate intake of micronutrients such as folate, potassium, magnesium, vitamins A and C, and B vitamins. Dairy product avoidance is also common, especially during periods of disease activity [16]. Without adequate alternate sources of calcium, this can be especially detrimental to patients with IBD given the increased risk of osteoporosis and osteopenia associated with corticosteroid treatment and prolonged inflammation [66].

Under the guidance of a professional, the risk of inadequate nutrient intake due to dietary modification can be mitigated by the provision of education around alternate sources of key nutrients, or supplemental nutrition where appropriate. However, patient accessibility to reputable sources of dietary information and advice can be limited, a challenge that can be hampered by medical practitioner scepticism around the utility of dietary therapy as an adjunct to drug therapy. For instance, a survey of 414 UK and NZ gastroenterologists found views of the efficacy of dietary exclusion in IBD and IBS symptom management to be mixed and the majority referring less than 25% of these patients to dietetic services [67]. Consequently, patients seek dietary information and advice from other sources including websites, online forums and books, and fellow patients [33,39] that can lead to self-imposed dietary restrictions and an increased risk of inadequate nutrient intake.

In the current study, a high number of dietary elements were identified with symptom onset and exacerbation, specifically fruit, vegetables and dairy products. These results suggest that self-imposed dietary restrictions could contribute to the high rates of malnutrition seen in the IBD population, a concept recently reported by Lim, Kim, and Hong [58]. Health providers may be unaware of the extent of patient views regarding diet and symptoms and should be encouraged to offer their patients nutritional counselling not only soon after diagnosis but also when attempts to suppress ongoing disease activity are ineffective. Besides ensuring patients understand and are able to implement the principles of a balanced diet, any dietary modification could be appropriately monitored and the risk of inadequate nutrient intake reduced.

While the success of available exclusion diets is limited, we may be overlooking the potential of an antigen-response based exclusion diet. Patients with IBD have been shown to have significantly higher rates of food hypersensitivity (allergies and intolerances), and encouraging results have been observed in response to IgG or IgE-titre based exclusion diets [68,69,70,71,72,73,74,75]. Further research in this area using objective measures of disease activity is needed to determine if the intake of specific dietary antigens correlates directly with IBD symptoms.

It is irrefutable that a greater understanding of the role of diet in symptom-based disease behaviour is needed. Well-designed prospective trials are needed to investigate how the immune system of patients with IBD responds to different foods, specifically, whether foods alter intestinal permeability, gene expression, or microbiota composition. Improved evidence could lead to development of evidence-based diet recommendations, a necessary tool for the improved management of IBD. Dietary approaches could also provide a low risk alternative for when drug therapy is responsible for severe side-affects or has lost its effectiveness.

### 4.8. Limitations

The study design was observational and thus causality cannot be ascertained. Although participants were asked to indicate dietary elements they associate with their symptoms, some may have been identified due to personal preference or as a result of information sought or advice received, rather than a perceived association with symptoms. Further, recall bias is an inherent and unavoidable limitation associated with retrospective dietary studies and could have influenced the study findings.

The prevalence of IBS in IBD patients is 35–44% [45], and the odds ratio of IBS in women compared with men is 1.67 [76]. Given the predominance of female participants (71%) and difficulties distinguishing between IBD and IBS symptoms [77], some participants may have mistakenly reported associations between dietary elements and IBS symptoms. As the baseline prevalence of IBS in this cohort was not determined, potential differences between participants affected and not affected by IBS were unable to be investigated.

The questionnaire did not distinguish between foods that can be consumed both raw and cooked, in some cases limiting the ability to deduce the primary symptom effect. Disease phenotype data were not collected, preventing the analysis of possible differences in effects dependent upon disease location and extent, and stricturing or penetrating disease in participants with CD. Finally, the study did not have a target ratio of patients with CD and UC, resulting in almost double the number of patients with CD. While this is consistent with recent patterns of IBD diagnosis in NZ [78,79], the smaller number of participants with UC may have resulted in this disease subtype being underrepresented.

## 5. Conclusions

This study showed that patients with IBD strongly believe diet can affect the duration and severity of their symptoms. The dietary elements most frequently identified with symptom onset and exacerbation are predominantly high in fibre, dairy-based or cooked by deep frying; while symptom reduction was associated with distinct dietary elements including banana, rice, white bread, and white meat. Further research is required to determine the extent of self-imposed symptom-associated dietary restriction in this patient group, and the subsequent effect on nutrient status.

## Figures and Tables

**Table 1 nutrients-12-02975-t001:** Characteristics of 233 participants.

Characteristics		All IBD	CD	UC	IBDU	*p* Value
Disease subtype *n* (%)			146 (63)	75 (32)	12 (5)	
Gender *n* (%)	Female	165 (71)	106 (73)	51 (68)	8 (67)	0.736
Age (years) mean ± SD		40.8 ± 14.9 *n* = 230	40.0 ± 14.9 *n* = 145	41.0 ± 14.5 *n* = 73	49.5 ± 15.4	0.105
Diagnosis age (years) mean ± SD		29.7 ± 13.1 *n* = 225	28.3 ± 12.4 *n* = 142	31.1 ± 13.2 *n* = 71	38.8 ± 17.4	0.015 *
Diagnosis age (years) *n* (%)	<20 20–40 >40	49 (22) 125 (55) 51 (23) *n* = 225	37 (26) 76 (54) 29 (20) *n* = 142	10 (14) 44 (62) 17 (24) *n* = 71	2 (16) 5 (42) 5 (42)	0.101 0.375 0.029 *
Disease duration (years) mean ± SD		11.0 ± 10.1 *n* = 228	11.6 ± 10.1 *n* = 143	9.9 ± 10.5 *n* = 73	10.7 ± 6.7	0.510
Disease duration (years) *n* (%)	<11 11–20 >20	144 (63) 50 (22) 34 (15) *n* = 228	83 (58) 38 (27) 22 (15) *n* = 143	53 (73) 10 (14) 10 (14) *n* = 73	8 (67) 2 (17) 2 (17)	0.171 0.635 0.438
Family history of IBD *n* (%)	Yes	65 (29) *n* = 228	47 (33) *n* = 144	17 (24) *n* = 72	1 (8)	0.108
Active disease in the last 12 months, *n* (%)	Yes	177 (79) *n* = 227	109 (77) *n* = 142	60 (82) *n* = 73	8 (67)	0.412

IBD = inflammatory bowel disease; CD = Crohn’s disease; UC = ulcerative colitis, IBDU = inflammatory bowel disease unclassified. Statistical analyses were conducted by Chi-square test for categorical variables and by 1-way ANOVA for continuous variables. * Significant difference between CD and IBDU (*p* < 0.05).

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
