# Peer review of "Inflammatory Bowel Disease: Are Symptoms and Diet Linked?"

_nutrients, 2020, doi:10.3390/nu12102975_

Round 1
Reviewer 1 Report
The issue of diet is probably most commonly raised subject by people living with IBD. In that respect this manuscript describing dietary patterns and beliefs presenting experience fromNew Zealand, addresses an emotive and important subject. There are , however, now a number of studies addressing this in IBD patients in general and in that respect the study lacks novelty.
Studies in IBD sub-groups are now being published (patients with "inactive" UC, and other sub-groups) and are now needed.
Please see my comments below:
- Your introduction could be shortened by a paragraph presenting most relevant information to set the scene for your study.
- You have provided your aims, objectives and conclusions also in the introductory paragraph which are misplaced there.
- You have included ALL IBD patients but have not defined disease phenotype or activity. This is a significant limitation as people with clinically active disease may have different dietary habits to those with active IBD. Further ore, those with stricturing or penetrating disease may have lower tolerance for example to fibre.
- How did you ensure that patients with IBD were confirmed to have the disease before they were recruited? Presenting advertisements in support groups also adds selection bias with more motivated or potentially refractory patients being more likely to form a part of these organisation.
- Did you not offer the questionnaire to consecutive patients attending the clinic and verify the diagnosis?A denominator showing how many patients were screened and how many willing to answer many demonstrate need and willingness to consider answering a diet related questionnaire and also help but defining disease activity even if using clinical scoring systems only.
- What was the basis of selecting those who had lived in New Zealand predominantly in the previous 2 years?
- Using self-reported disease activity in the last 12 months can be misleading as patients who have had a varying disease course in that year could easily mis-classify their disease.
- A table showing the questionnaire you used should be made available as a supplementary figure.
- How did you establish "IBDU" in those completing an on line version of the questionnaire?
- What do you mean by"lost-to follow up"? In a questionnaire study, one would assume they answered some or most of the questionnaire. There was no follow-up questionnaire. Lost to follow up is difficult to contextualise.
- The female predominance might suggest an IBS overlay given the high prevalence reported in females for IBS and similar symptoms.Please comment and acknowledge in limitations.
- Do you know the baseline prevalence of IBS in this cohort?
- How was "IBD-onset" determined? Did it mean onset of disease or symptoms? If onset of disease as is typically implied, there is a significant element of recall bias which should be acknowledged in the discussion.
- Recall bias would have significantly limited your findings as is true with most dietary studies not to mention the complex interactions within food and food groups which are difficult to tease out. This unavoidable limitation needs to be mentioned.
- In your discussion, you state that dietary restrictions account for high rate of malnutrition.Your study neither sought to study nor demonstrates this. If this is a comment based on previous literature, it needs to be referenced.
- A recent study by Crooks B et al. "The Dietary Practices and Beliefs of People living with Inactive Ulcerative Colitis" European Journal of Gastroenterology and Hepatology could be referenced and your findings where relevant contrasted in the discussion.
Reviewer 2 Report
The influence of the diverse components of the diet, in the onset, maintenance, exacerbations and remission of the IBD is very relevant to know because is related in more than 50% of patients with these diseases including CD and UC
In this paper the authors have analyzed several components of the diet and their influences of all these events in the clinical course of these patients and describe interesting relationships obtained through observational questionnaires extracted directly form the experience of a large number of affected patients
An interesting new aspect is the way of cooking or preparing the foods that can also influence in their deleterious or beneficial effects
The study is very interesting and well done
I have included some minor suggestions :
Review.Nutrients : 937697
Q1 : Title
Is OK
Q2 : Abstract
In line 13, would be especified incidence or prevalence and all over the world or other comparison with another countries
In line 22, it must have added after the figures “in all IBD participants”
In the keywords, the two last words are synonimus; one of them, must be eliminated
Q3 : Introduction
Line 41 : Please write resection, not re-section
Line 42 : Please add, corticosteroids and biologics
Line 81 : Please write among, not amongst
Line 84 : Please divide this paragraph in two parts, beginning the last one, by the objectives of this study……
General recommendation . There is a great number of references included in this part (43 in total). It would be suggested to move some to the Discussion part in orde to account in this part only 20-25 in total.
Q4 : Material and Methods
2.1. Participants
It would be convenient to specify the diagnostic criteria used for confirming the diagnosis of CD, UC and IBDU
2.2. Questionnaire
It would be better to write in number the 7 additives and the 4 cooking methods
2.3. There were inform writing consent of all participants?
2.4. The complete name of JASP system, must be included
Q5. Results.
- 1.
Please specify, which are the reasons to have included the double of CD (142), than UC pts. (71)?.
And the mean age around 40 years is some elevated?
3.2. Symptom onset
It would be better to write in plural. Symptoms, instead symptom
3.3. Symptom Exacerbation
It would be better to write in plural. Symptoms, instead symptom
3.4 . Symptom Reduction
It would be better to write in plural. Symptoms, instead symptom
3.5. Other Responses
Is OK
3.6. FODMAPS
Is OK
- Discussion
4.1. Symptom Onset and Exacerbation
Is OK
4.2. Fruit and Vegetables
Is OK
4.3. Dairy Products
Is OK
4.4. Gluten
Is OK and very well explained
4.5.
Is OK
4.6. Symptom Reduction
It would be better to write in plural. Symptoms, instead symptom
The last paragraph of this section is very interesting explaining that the response to the dietary elements is often working in a heterogeneous fashion in some patients in CD and UC also with contradictory effects.
4.7. Dietary Modification
The consequences of deprivation diets on the micronutrients is very relevant
The importance to have a nutritionist in the management of IBD patients is clearly remarked
The final conclusions are clearly exposed
4.8. Limitations
I fully agree with the limitations described and also can de included the need to eliminate the IBDU because is very short (only 12 people)
The Tables are good enough and I think they must be included into the manuscript
5.References
They are in good number and very well selected
Round 2
Reviewer 1 Report
Thank you for revisions your manuscript in line with our recommendations.
Whilst i do think that self reporting of IBD symptoms in the last 12 months can lead to a huge recall bias and indeed questionable with reference to what constitutes IBD symptoms (some overlap with IBS as pointed out in my comments earlier), you have acknowledged this in your limitations
The paper I referred to from the European Journal of Gastroenterology ( is published one Pubmed and should be cited as below:
Crooks B, McLaughlin J, Matsuoka K et al.The dietary practices and beliefs of people living with inactive ulcerative colitis Eur J Gastroenterol Hepatol. 2020 Sep 17. doi: 10.1097/MEG.0000000000001911
It is probably the first to look at prevalence of food related symptoms in inactive UC.
